# Osteopontin Splicing Isoforms Contribute to Endometriotic Proliferation, Migration, and Epithelial-Mesenchymal Transition in Endometrial Epithelial Cells

**DOI:** 10.3390/ijms232315328

**Published:** 2022-12-05

**Authors:** Nguyen-Tuong Ho, Shu-Wei Lin, Yi-Rong Lee, Chii-Ruey Tzeng, Shu-Huei Kao

**Affiliations:** 1International Ph.D. Program in Medicine, College of Medicine, Taipei Medical University, Taipei 11031, Taiwan; 2School of Medical Laboratory Science and Biotechnology, College of Medical Science and Technology, Taipei Medical University, Taipei 11031, Taiwan; 3Taipei Fertility Center, Taipei 11073, Taiwan; 4Department of Obstetrics and Gynecology, School of Medicine, College of Medicine, Taipei Medical University, Taipei 11031, Taiwan; 5Ph.D. Program in Medical Biotechnology, College of Medical Science and Technology, Taipei Medical University, Taipei 11031, Taiwan; 6Center for Reproductive Medicine, Department of Obstetrics and Gynecology, Taipei Medical University Hospital, Taipei 11031, Taiwan

**Keywords:** endometriosis, osteopontin splicing variants, CD44, αvβ3, migration, proliferation, epithelial-mesenchymal transition

## Abstract

Osteopontin (OPN) isoforms, including OPNb and OPNc, promote malignancy and may contribute to the pathogenesis of endometriosis, a benign disorder with multiple characteristics resembling malignant tumors. In our experiments, OPNb and OPNc were significantly overexpressed in both endometriosis and adenomyosis compared to the normal endometrium. Upregulation of CD44v and the epithelial–mesenchymal transition (EMT) process was also present in endometriotic lesions. Overexpression of OPNb and OPNc splicing variants in endometriotic cells evoked morphological changes, actin remodeling, cell proliferation, cell migration, and EMT through binding OPN ligand receptors CD44 and αvβ3, subsequently activating the PI3K and NF-ĸB pathways. We elucidated the causal role of OPN splice variants in regulating endometriotic cell growth, which may promote the development of OPN-targeted therapies for patients suffering from endometriotic disorders.

## 1. Introduction

Endometriosis is the presence of ectopic endometrial tissue comprising both endometrial stromal cells (ESCs) and endometrial epithelial cells (EECs) at a location other than the uterine cavity. While endometriosis is not malignant, it has tumor-like characteristics, which enable endometriotic lesions to approach and penetrate the surrounding tissue, thus precipitating anatomical and functional changes in affected organs and subsequently causing irreversible disorders, including depleted ovarian reserve and infertility [1].

While the multifactorial pathology of endometriosis is not fully understood, endometriotic cells have been shown to express molecular characteristics similar to cancer cells, including invasion, migration, and proliferation. Epithelial–mesenchymal transition (EMT), a complex and reversible transformation of endometrial epithelial cells, downregulates cellular polarization and cell-cell adhesion by disassembling adherens junctions, thus allowing the transformed cells to invade the basement membrane and peritumoral stroma and metastasize [2]. Epithelial cells undergoing EMT subsequently achieve mesenchymal characteristics, including migratory or invasive potential, apoptosis resistance, and increased extracellular matrix components. During EMT, E-cadherin, the hallmark associated with this phenomenon, substantially decreases in association with a remarkable increase in N-cadherin and vimentin, which distinctively characterizes the cellular mesenchymal phenotype [3,4]. The EMT process also exists in endometriosis, although the exact mechanism has yet to be fully elucidated [5].

The CD44 receptor, a glycosylated membrane receptor associated with tumor chemoresistance and metastasis [6], also plays a role in endometriosis development [7]. The overexpression of CD44 variants activates a diversity of downstream pathways, including CD44/EGFR/PI3K-Akt and CD44/NF-κB, thus characterizing endometriosis with dominant migration and invasiveness [8,9]. The expression of CD44 promotes EMT by activating the CD44/EGFR/PI3K-Akt and CD44/NF-κB signaling pathways in carcinoma [6,10,11,12]. Nevertheless, it remains unclear how CD44 interacts with EMT in patients with endometriosis.

Osteopontin (OPN), a multifunctional cytokine that results from phosphorylation, contributes to the pathogenesis of various malignant conditions through different signaling pathways, including the EGFR/ERK/MMP-9 and CD44/Akt/NF-ĸB pathways; furthermore, OPN promotes cell survival, proliferation, and invasiveness in conjunction with tumor angiogenesis [13]. In addition, OPN interacts with CD44 in a ligand-receptor relationship, inducing cellular migration and growth [14]. Alternative splicing has been reported in OPN biosynthesis, with three different isoforms identified: OPNa (full-length OPN), OPNb, and the shortest, OPNc [15]. Regarding OPN function in oncogenesis, different OPN variants display specific functions and biological processes based on the particular host cell types. Both the “full-length” OPNa and OPNc precipitate tumoral invasiveness in human glioma cells [16]. OPNb and OPNc were recognized as pro-oncogenic factors that trigger various tumor progression pathways in prostate carcinoma, thereby mediating resistance as well as responses to chemotherapy-induced apoptosis [17]. The presence of OPN in breast and ovarian cancer cells reveals a bifunctional characteristic: OPNc supports tumoral progressiveness and becomes a particular marker for transformed cells, while OPNa promotes cell adhesion and therefore inhibits cellular dissemination [18,19]. OPN exerted either an advantageous or deleterious effect on a variety of both malignant and chronic diseases in a tissue-specific manner. The diversity in the expression of different OPN isoforms could serve as a foundational prognostic factor and potentially aid in decision-making for cancer treatment [18,20]. The ligation between osteopontin and different CD44 variants (i.e., CD44v6 or CD44v9) was reported to facilitate tumor migration and metastasis [21], and EMT may play a role in this relationship [22]. Further research is needed to determine how different OPN isoforms interact with CD44 and its variants.

Previous studies reported the expression of OPN among patients with endometriosis. In combination with a high serum level of OPN, increased local expression of this protein in endometrioma cells is correlated with cellular migration, invasion, and proliferation [23] and has the potential to be a promising marker of endometriosis [24]. OPN knockdown in ectopic ESCs inhibits cell migration and invasion by suppressing the uPA/PI3k/Akt pathway [25]. In HEC1A, an endometrial cancer cell line widely considered to be a good candidate for the in vitro model of the human endometrium, overexpression of OPN was found to be positively correlated with EMT-related factors through mediating the AKT and ERK1/2 signaling pathways [26]. At the eutopic endometrium site, OPN expression was reported to be significantly lower than that in corresponding eutopic tissue and endometrium from women without endometriosis, especially during the implantation window [27]. CD44 and its splicing variants are also correlated with the development of early endometriotic lesions [7]. However, the interaction between OPN and CD44 in endometriosis remains poorly understood.

In this study, we examined whether OPN interacted with the CD44 receptor or integrin αvβ3 and the distinctive function of each OPN splicing variant in the pathogenesis of endometriosis by using an in vitro endometriosis model with HEC1A cell line. The results could promote the development of OPN-directed therapies for women with endometriosis and shed light on how endometriosis has negative effects on female reproduction.

## 2. Results

### 2.1. Expression of OPN Isoforms, CD44 Variants, and EMT Markers in the Different Endometriotic Lesions

Tissues including adenomyotic lesions, ovarian endometriotic fluids, and endometria from patients with uterine myoma were collected for real-time PCR and protein quantification, while OPN levels in serous and peritoneal fluid samples were measured using an enzyme-linked immunosorbent assay kit. OPN levels in the sera of women with ovarian endometriosis were greater than those from non-endometriotic women (*p* < 0.05). The concentration of OPN in peritoneal fluid was also high but significantly reduced after treatment with a gonadotropin-releasing hormone agonist (GnRHa) (*p* < 0.05) (Figure 1A). Furthermore, the representative immunoblots revealed a significant increase in OPN protein expression in the endometriotic lesions compared to the myoma control group: the highest level of OPN was found in the adenomyotic tissues (*p* = 0.001), followed by the ovarian endometrioma (*p* = 0.03) (Figure 1B).

To examine the profiling of OPN isoforms, PCR and reverse transcription-PCR were used to analyze the mRNA levels of OPN splice isoforms in the specimens. Quantitative analysis of the mRNA levels of OPN splice isoforms by real-time quantitative PCR (qPCR) is shown in Figure 1C. The electrophoretogram highlighted that the expression levels of the OPNb and OPNc isoforms in adenomyosis were more obvious than those in the control group (*p* < 0.001). OPNb was also found to be significantly overexpressed in ovarian endometrioma (*p* = 0.01) compared to eutopic endometrial tissue from patients with fibroids. In quantitative analysis, we found that OPNc, which has been reported to be increased in endometrioma tissue, was only significantly higher in the specimens of patients with adenomyosis (Figure 1D). Both OPNb and OPNc displayed a significantly positive correlation with serum cancer antigen 125 (CA-125) (*p* < 0.05) (Figure 1E).

In addition, a remarkable change in the distribution of CD44 isoforms in the endometriotic tissues was observed. Specifically, the dominant expression of CD44s in the normal eutopic endometrium was altered to an increased level of CD44v in both the endometrioma and the adenomyosis tissues (*p* < 0.05), accompanied by a decreased level of the standard counterpart CD44s (Figure 2A). CD44v upregulation was also found in qPCR results after normalization with β-actin (Figure 2B–D). In both ovarian endometrioma and adenomyosis tissues, CD44v upregulation was positively correlated with OPNb and OPNc expression (*p* < 0.05). Additionally, CD44s was found only to be correlated with OPN isoforms in adenomyotic lesions (*p* < 0.05) (Figure 2D).

The EMT process was prominent in ectopic endometrial tissue: Vimentin and N-cadherin levels increased in both adenomyosis and endometrioma, while a decrease in the epithelial marker E-cadherin was found only in adenomyotic tissue (*p* < 0.05) (Figure 3A). Electrophoretogram and qPCR analysis highlighted the overexpression of N-cadherin and Vimentin in endometriotic cells, while the corresponding suppression of E-cadherin was also observed (Figure 3B,C). The cell adhesion molecules ICAM and VCAM were used as the biomarkers for endometriosis and contribute to EMT progression. Both adenomyosis and endometrioma tissues had elevated ICAM and VCAM mRNA expression levels.

### 2.2. OPN Variants Regulated Epithelial-Mesenchymal Transaction by Inducing Actin Skeleton Remodeling

HEC1A cells were differentially transfected with an empty vector or vectors containing OPNb and OPNc. The electrophoretogram for identifying the expression of OPN isoforms in transfected cells is shown in Figure 4B,C. The overexpressed cells were then knocked down by siRNA-OPN or siRNA-CD44, with the efficiency depicted in Figure 4D,E.

Phalloidin with tetramethylrhodamine (TRITC) and DAPI were used to stain the intracellular cytoskeleton protein actin and nucleus, respectively. OPN isoforms were found to promote actin cytoskeleton remodeling inside endometriotic cells, exhibited by elongated cell shapes, numerous pseudopodia, and redistribution of β-actin to pseudopodial tips. Additionally, the malformation of these pseudopodia and actin remodeling were diminished under the effect of OPN siRNA, CD44v siRNA, or αvβ3 integrin inhibitor (Figure 5A). The EMT process was proven to be stimulated through the activation of the standard form CD44s and variant form CD44v receptors. Both OPNb and OPNc significantly increased CD44 activities, but the increased expression of CD44 family receptors was most enhanced by OPNb (Figure 5C). The EMT process was provoked with upregulation of the mesenchymal markers Vimentin and N-cadherin, accompanied by a reduction in the epithelial marker E-cadherin (*p* < 0.05). Silencing OPN overexpression using OPN-siRNA inhibited the EMT process, as indicated by N-cadherin and Vimentin downregulation (Figure 5B,C).

Moreover, the scratch wound assay revealed the dominant cell migratory ability in OPNc-overexpressing cells (*p* = 0.001), and this effect was, to a lesser extent, also promoted in the OPNb-transfected cells (*p* = 0.003). Silencing the expression of OPN partially reduced the migratory ability (*p* < 0.05). Furthermore, the suppression of OPN ligand-receptor binding, including CD44 and αvβ3 integrin, significantly deteriorated the migration ability in OPN-overexpressing endometriotic cells (Figure 6A,B).

### 2.3. OPN Splice Variants Promoted Cell Proliferation and Migration by Binding to αvβ3 Integrin and CD44 Receptors in Endometriotic Cells

In OPNb- and OPNc-overexpressing HEC1A cells, the cellular morphology was obviously altered to be more irregular and elongated than that in the original nontransfected cells. However, after incubation with OPN siRNA, CD44v siRNA, or αvβ3 integrin inhibitor (1 µg/mL), the transformed cells retrieved the original form identical to the control cells (Figure 7A). Additionally, a more intense increase in confluency of the OPN-overexpressing cells implied the promoted potential for cell proliferation, which was also attenuated after treatment with OPN siRNA, CD44v siRNA, or αvβ3 integrin inhibitor (anti-αvβ3 antibody).

A bromodeoxyuridine cell proliferation assay was used to determine the effect of OPN isoforms on cell proliferation. Both OPN isoforms promoted higher proliferation compared to the control group (*p* < 0.05). However, the OPNb overexpressed cells had a significantly higher proliferation rate than the OPNc overexpressed (*p* = 0.003) and the control vector-only HEC1A cells (*p* = 0.001) (Figure 7B).

### 2.4. OPN Splice Variants Modulated the Migration of Endometriotic Epithelial Cells through the PI-3k or NF-ĸB Signaling Pathways

To explore whether the nuclear factor kappa-light-chain-enhancer of activated B cells (NF-κB) and phosphoinositide 3-kinase (PI-3K) signaling pathway were involved in the pathogenesis of endometriosis, endometriotic cells were treated with 10 µM wortmannin, a selective and irreversible PI3K inhibitor, or CAPE (caffeic acid phenethyl ester), an antioxidant inhibiting NFκB activation. After 48 h of incubation with wortmannin and CAPE, endometriotic cells overexpressing OPN isoforms were found to have a significant reduction in migratory ability (Figure 8A,B) and proliferation potential (Figure 8C) (*p* < 0.05). Wortmannin and CAPE ameliorated NF-ĸB translocation into the nucleus, while OPN suppression nearly eliminated NF-ĸB activation (Figure 8D).

## 3. Discussion

Endometriosis is common among women of reproductive age and is characterized by a high recurrence rate and persistent complications to their fecundity. Albeit benign, the development of endometriosis exhibits a range of characteristics resembling tumorigenesis, such as cell proliferation, migration, and invasion. Studies have focused on the discovery of endometriosis biomarkers or targeted molecular therapies, but there is no consensus regarding optimal strategies for managing endometriosis.

OPN is a versatile extracellular structural glycoprotein that also binds to diverse receptors, contributing to various biological and metabolic processes. Overexpression of OPN is advantageous for wound recovery, bone homeostasis, and extracellular matrix function [28,29,30], but it was suggested to be deleterious in the pathogenesis of malignant transformation and cancer, with crucial contributions to the immune response, proliferation, adhesion, migration, and invasion of tumor cells [31,32]. High OPN expression has been reported during embryo implantation and following pregnancy in both in vitro [33,34] and in vivo experiments [35,36], demonstrating its crucial role in the conception process and maintenance of pregnancy. In contrast, Konno et al. first investigated the role of OPN in endometriosis and reported abundant amounts of OPN in endometriosis tissue through immunohistostaining [37], suggesting that OPN participated in the establishment of endometriosis. Another study reported a remarkably higher serum OPN level in women with endometriosis compared to nonendometriosis controls [38], which was in line with our findings. GnRHa has been proved to be effective in endometriosis treatment and prevention of post-operative recurrence [39]. Our experiment demonstrated a significantly decreased OPN concentration after GnRHa treatment, suggesting that OPN could serve as a potential biomarker for endometriosis diagnosis and treatment follow-up.

Additionally, we revealed a range of OPN concentrations in different endometriotic lesions, with the highest OPN expression seen in adenomyotic tissues. Adenomyosis shares similar developmental features with endometriosis but is not simply endometriosis of the uterus. While Sampson’s retrograde menstruation theory is widely accepted and provides the most robust evidence for endometriosis development, there are essentially two major and competing explanations for the pathogenesis of adenomyosis: invagination and metaplasia [40]. The invagination hypothesis is based largely on the tissue injury and repair (TIAR) theory proposed by Leyendecker et al. [41], in which tissue autotraumatization caused by malfunction of the junctional zone (JZ) initiates the TIAR process through activated estrogen receptors [42]. Genes and pathways involved in the regulation of apoptosis, hormone responsiveness, and extracellular matrix remodeling were found to be upregulated in adenomyotic lesions. The remarkably higher expression of OPN in adenomyosis compared to others was a reasonable finding, as OPN was also found to be strongly correlated with extracellular matrix deposition in the tissue recovery process [30]. Moreover, the expression levels of OPNb and OPNc in adenomyosis were both higher than those in nonuterine endometriosis and nonendometriosis women (*p* < 0.001). OPNb and OPNc were also positively correlated with serum CA-125 levels, a well-known biological marker of endometriosis severity [43], indicating that OPN or OPN isoforms could serve as prognostic factors for women suffering from this chronic systemic disease. The different expression of OPN and its splice variants in uterine endometriotic lesions compared to other sites probably results from the variety of the pathogenesis and requires further investigation.

Alternative splicing, a process during gene expression that allows a single gene to code for multiple proteins, occurs in over 90 percent of protein-coding genes and contributes to the pathophysiology of many diseases [44]. Three isoforms of OPN have been reported: the typical full-length OPNa with 7 exons, the OPNb without exon 5, and the shortest, OPN-c, lacking exon 4. The contribution of OPN and its isoform (OPNb, OPNc) to tumorigenesis has been reported by many studies [15,19,45,46]. Interestingly, OPNb and OPNc variants not only share common characteristics but also contribute separately to disease development due to their different missing exons. In gastric cancer, overexpression of OPNb strongly induces cell survival by regulating Bcl-2 family proteins and CD44v expression, while OPNc most significantly promotes cell migration and invasion by increasing the secretion of matrix metalloproteinase-2, urokinase-type plasminogen activator, and interleukin-8 [47]. In esophageal adenocarcinoma, OPNb cells exhibit significantly increased cell adhesion, while OPNc cells show enhanced cell detachment [48]. In ovarian cancer, OPNc, but not OPNb, contributes to cancer progression by modulating cancer cell viability, plasticity, and cisplatin resistance [19,49]. The pathogenesis of endometriosis is complicated and possesses multiple characteristics typically attributed to tumors. Despite a number of studies reporting the relationship between OPN and endometriosis development, the influence of OPN splice variants on this gynecological disorder remains unclear. Our data revealed that both OPNc and OPNb contributed to endometriosis pathogenesis, including EMT, proliferation and migration. HEC1A cells transfected with OPNb/OPNc were morphologically transformed from a round shape to a slender phenotype similar to mesenchymal cells, along with increased actin remodeling and pseudopodia formation. This phenomenon represented a rise in EMT activity, which could induce disease development by enhancing proliferation and migration. However, each splice variant also exhibited a distinct profile in the progression of endometriosis. In particular, while OPNb dominantly promoted cell proliferation, the selective overexpression of OPNc was associated with remarkably greater migratory ability. This finding implies that each OPN splicing variant has a distinct and detrimental effect on endometriosis progression, and downregulation of both OPNb and OPNc may inhibit the lesion features evoked by these isoforms, including morphological changes, actin remodeling, proliferation, migratory ability, and EMT process.

OPN was also found to be a protein ligand of the glycoprotein CD44. CD44 is a cell surface receptor that interferes with cell-cell and cell–matrix interactions [50]. Studies have shown that upregulation of CD44, together with other adhesion molecules, initiates a cascade of events and modulates adhesiveness, motility, extracellular matrix degradation, cell proliferation, and survival [51]. To date, CD44s has been the most frequently reported isoform in malignancy, but other CD44 variants (e.g., CD44v) were also found to regulate redox homeostasis and promote neoplasia in several cancer types [52,53]. CD44 expression and the interaction between CD44 and OPN isoforms in the pathogenesis of endometriosis have been examined in several studies, but the results have been inconsistent [7]. Our study is the first to report a change from the dominant expression profile of CD44s in the normal endometrium to that of the CD44v isoform in ectopic endometriotic tissues. Similar to silencing OPN isoforms, knockdown of the expression of CD44 in both OPNb- and OPNc-overexpressing HEC1A cells reversed the cellular shape alteration and significantly reduced EMT and cell migration. Another ligand receptor of OPN, αvβ3 integrin, has also been reported as a modulator of EMT and the tumorigenesis process [54]. The OPN-overexpressing endometriotic cells reverted to their original form after αVβ3 integrin inhibition. Incubation with anti-αVβ3 antibody also reduced both cellular migratory ability and proliferative potential. In summary, our study revealed that overexpression of OPNb and OPNc could regulate the EMT, proliferation, and migration of endometriotic cells by activating CD44 or αvβ3 integrin receptors.

Previous studies have reported that OPN regulates the NF-κB or PI3K/Akt pathway by binding to integrin-αvβ3 or CD44, subsequently inducing proteolytic enzymes such as matrix metalloproteinases or urokinase-type cytoplasm [55,56]. By interacting with the CD44 family of receptors, OPN can activate the cell survival (anti-apoptosis) signals through the hospholipase C-γ (PLCγ–protein kinase C (PKC)–phosphatidylinositol 3-kinase (PI3K)–Akt pathway. OPN also exerts its function by activation of hypoxia-inducible factor-1 alpha (HIF-1α) pathways via the PI3k/AKT pathway, which leads to enhanced tumor cell survival, proliferation, invasion, EMT, and angiogenesis. Upon binding to αvβ3, OPN can promote cell motility and tumor progression through nuclear factor-inducing kinase (NIK)–ERK (extracellular signal-related kinase) and MEKK1 (mitogen-activated protein kinase kinase kinase1)–JNK1 (c-Jun N-terminal kinase 1) signaling pathways to active AP1 [13]. Concurrently, OPN can transactivate epidermal growth factor receptor (EGFR) to trigger ERK phosphorylation, which eventually leads to activation of AP1. AP1 and NF-kB can activate both metalloproteinase (MMP)-2 and MMP-9 to promote cell movement and metastasis [57,58]. Moreover, the crosstalk between the PI3K/Akt and NF-κB pathways has also been previously implied [59,60,61]. In this study, we demonstrated that both OPNb and OPNc affected cell migration through the NF-κB and PI3K/Akt pathways. Inhibition of the NF-κB pathway with CAPE or the PI3K pathway with wortmannin reduced the invasive potential of endometriotic cells, including proliferation and migration. OPN isoforms could play the role of a cross-talk or an independent factor modulating both the PI3K/Akt and NF-κB pathways, thereby mediating the pathogenesis of endometriosis or adenomyosis.

In conclusion, our study revealed the interaction between OPN isoforms and OPN ligand receptors and their influence on the progression of endometriosis using an endometrial epithelial cell model. OPNb and OPNc were overexpressed in endometriotic tissues and modulated cellular morphology, actin remodeling, proliferation, and migration. In endometrial epithelial cells, the regulation of migratory ability was performed by both OPN splicing variants through binding to CD44 and αvβ3 integrin receptors, subsequently activating the NF-κB and PI3K/Akt pathways (Figure 9). Further in vitro and in vivo studies are required to investigate the underlying mechanism of the interaction between OPN isoforms and the EMT process and cell proliferation, as well as to establish new OPN-targeted therapies for endometriosis.

## 4. Materials and Methods

### 4.1. Sample Collection

The specimen collection procedure was performed in accordance with the regulations of the Office of Human Research at Taipei Medical University (No. P950046). The inclusion criteria were: (i) normal ovulatory women > 20 years old; (ii) undergoing a scheduled surgery for uterine myoma, adenomyosis, or endometriosis; (iii) who agreed to provide samples and sign in the informed consent. Women who were at menopause or perimenopause, suffering malignant diseases or untreated anemia, or non-Taiwanese were excluded from this study. Eutopic and ectopic endometrial tissues from patients with myoma (n = 12), adenomyosis (n = 12), and ovarian endometriosis (n = 12) were collected at the Reproductive Medicine Center, Taipei Medical University Hospital, Taipei, Taiwan, as listed in Appendix A. All solid samples were washed with phosphate-buffered saline (PBS) three times and stored at −80 °C afterward. Additionally, we pre-operatively collected the blood samples from women with uterine myoma (n = 8) and ovarian endometrioma(n = 8). Peritoneal fluids were aspirated from infertile women with ovarian endometrioma (n = 8) during laparoscopic cystectomy and subsequent oocyte retrieval cycle using a long GnRHa protocol. The body fluids were frozen and stored at −80 °C for further use.

### 4.2. ELISA for OPN Levels in Body Fluids

The levels of OPN in the body fluids were measured with an enzyme-linked immunosorbent assay (ELISA) from Sigma-Aldrich (Merck, KGaA, Darmstadt, Germany). The assays were performed according to the instructions in the product data sheet. The results were collected and calculated according to a standard curve with standards of osteopontin provided by the manufacturer.

### 4.3. Endometriosis Cell Culture

HEC1A (ATCC: HTB-112TM) and human endometrial cells derived from clinical samples were used in this study. Both were cultured in phenol red-free Dulbecco’s Modified Eagle’s Medium/Nutrient F-12 Ham (DMEM/F12, Thermo Fisher Scientific, Waltham, MA, USA) medium supplemented with 10% fetal bovine serum (FBS, Thermo Fisher Scientific), 100 U/mL penicillin G sodium, and 100 μg/mL streptomycin sulfate in a cell incubator (5% CO_2_ at 37 °C) before use.

### 4.4. Cell Transfection

HEC1A cells (5 × 10^5^) were prepared in 6-well plates and transfected for approximately 6 min. After that, the cells were washed with PBS before adding 1 mL OPTI-MEM (Thermo Fisher Scientific). The reaction lasted 5 min at room temperature with 3–5 µL Lipofectamine 2000 reagent and 500 mL OPTI-MEM. Then, 4 µL of pcDNA3.1 (OPNb), pDest490 OPN-ct (OPNc) vector, or pcDNA3.1 control vector was added, and the reaction was allowed to finish in 25 min. The solution was applied to each well and recovered after 4–5 h. Next, 1.5 mL of DMEM/F12 medium with 5% FBS was added. After culturing overnight, the medium was replaced, and the cells were incubated in complete medium for 48 h prior to treatment. The transfection efficiency was monitored by detecting the levels of OPNb and OPNc mRNA. Vector-transfected cells were collected and sequenced for efficacy evaluation.

### 4.5. Scratch Wound Assay

A scratch assay was used to measure cell migratory ability. Cells were cultured in DMEM/F12 with 1 µM mitomycin C to inhibit cell proliferation and allow the cells to form a confluent monolayer. A culture insert was used to create a discrete area of the confluent monolayer to form a cell-free zone into which the endometriotic cells at the edges of the wound could migrate. Wortmannin (PI-3K inhibitor, Merck KGaA, Darmstadt, Germany) and CAPE (NF-ĸB inhibitor, Merck KGaA) were used. The motion of cells in response to the test substance was observed under a microscope at various time points during culturing. The percentages of wound area were calculated in the indicated cells treated with OPN siRNA, CD44 siRNA, wortmannin and CAPE in the OPNb- or OPNc-transfected cells.

### 4.6. Protein Extraction and SDS-PAGE Western Blotting

A total of 10^6^ cells were cultured, and the cell pellet was collected, mixed with 100 µL protein lysis buffer, and then kept on ice for 30 min. The protein concentration was then quantified using Bradford’s method with Coomassie Brilliant Blue G-250 reagent (Thermo Fisher Scientific), with absorbance measured at a wavelength of 595 mm.

Aliquots of 50 µg purified protein were loaded onto a 10% SDS-PAGE gel and subjected to electrophoresis for 3 h before being transferred to a PVDF membrane at 4 °C for 100 min. The membrane was blocked with 5% skim milk powder in 0.1% Tris buffered saline with Tween 20 (TBST) buffer for 1 h before being incubated with a primary antibody (anti-OPN antibody, anti-GAPDH antibody, and/or anti-CD44 antibody) overnight. Bound primary antibodies were detected using the appropriate secondary antibody (horse anti-mouse IgG antibody). After each incubation, the membrane was washed with TBST 3 times for 10 min each time. Protein bands were detected using enhanced chemiluminescence (ECL).

### 4.7. RNA Extraction and RT-PCR

The cell pellet was lysed in 500 µL of nucleone and mixed with 200 µL of RNase-free water and then incubated for 5 min. The supernatant was then transferred to a new Eppendorf vial and precipitated for RNA isolation with the QIAGEN RNeasy Mini Kit (Hilden, Germany) as directed by the manufacturer’s instructions. The synthesized cDNA was then amplified using a mini-cycle TMPCR machine with 25–30 thermal cycles as described: 95 °C initial denaturation for 5 min, 95 °C denaturation for 40 s, 56 °C annealing for 40 s, and 72 °C primer extension. The final extension lasted for 5 min at 72 °C. PCR products underwent electrophoresis in 1.2% agarose gel with ethidium bromide (Sigma, St. Louis, MO, USA) at a concentration of 0.5 µg/mL. The gel was placed in a Mupid-2J mini tank (Cosmo Bio, Tokyo, Japan) containing 0.5% TBE buffer (0.1 M Tris, 0.1 M boric acid, 2 mM Na_2_EDTA, pH 8.0) before loading mixtures of PCR product and dye. The DNA samples were electrophoresed at 100 V for 30 min and checked for expression under UV light from a Viber Lourmat photoprint 008-SD machine (Cedex, France). The oligonucleotide sequences of primer pairs used herein are shown in Appendix A.

### 4.8. siRNA Transfection

Endometrial epithelial HEC1A cells were cultured until the confluency reached 50–60% for transfection. siRNAs with sense sequences OPN 5′-GGUCAAAAUCUAAGAAAGUUTT-3′ and CD44v 5′-CAUCGGAUUUGAGACCUGCAGGUAU-3′ (Ambion Silencer^®^, Lakewood, NJ, USA) (Appendix A) were suspended in 50 µL double-distilled water and transfected with Lipofectamine 2000 (Invitrogen, Carlsbad, CA, USA) according to the manufacturer’s protocol. The efficacy of transfection was evaluated using qPCR.

### 4.9. Laser Conjugate Microscopy

HEC1A cells with and without OPN overexpression were cultured on 0.17 mm coverslips until a confluence of 70–80% was achieved. The cells were then washed and incubated with fixation solution (4% formaldehyde and 0.03 M sucrose) for 1 h at room temperature, washed with PBS, permeabilized with Triton X-100 0.1% in PBS for 15 min, and then blocked with 5 mg/mL BSA for 1 h. After blocking, 75 µL phalloidin (5 U/mL, Sigma-Aldrich, Burlington, MA, USA) was added for 20 min, and then DAPI (1 µM) was added for 3 min. Finally, the slides were incubated with a small amount of anti-reducing agent VECTASHIELD^®^ mounting medium (Thermo Fisher Scientific). The prepared slides were then observed under a confocal microscope (TCS SPS AODS, Leica, Germany) to assess the distribution and remodeling of actin filaments in the various transfected cells.

### 4.10. Bromodeoxyuridine Cell Proliferation Assay

A total of 10^6^ HEC1A cells were seeded into each well of a 96-well plate and cultured at 37 °C, 5% CO_2_, and 95% humidity for 48 h before the experiment. The cells were then washed, treated with bromodeoxyuridine (BrdU) diluted 1:100 in 5% FBS/DMEM for 24 h, and fixed with FixDenat solution at 25 °C for 30 min. Cells were then consecutively treated with anti-BrdU (Merck KGaA), a mixed substrate component A and B solution, before being analyzed by an Invitrogen ELISA reader luminometer (Carlsbad, CA, USA).

### 4.11. Statistical Analysis

Student’s paired samples t-test was used for comparing two groups, and ANOVA with post hoc analysis was used for comparing data from ≥3 groups. The experimental results are expressed as the mean ± standard error of the mean (SEM). Linear regression analysis was used to determine the statistical significance of the relationship between the levels of OPN isoforms and serum CA-125 levels or between the levels of OPN isoforms and CD44 variants. A *p* value of less than 0.05 was considered statistically significant.

## Figures and Tables

**Figure 1 ijms-23-15328-f001:**
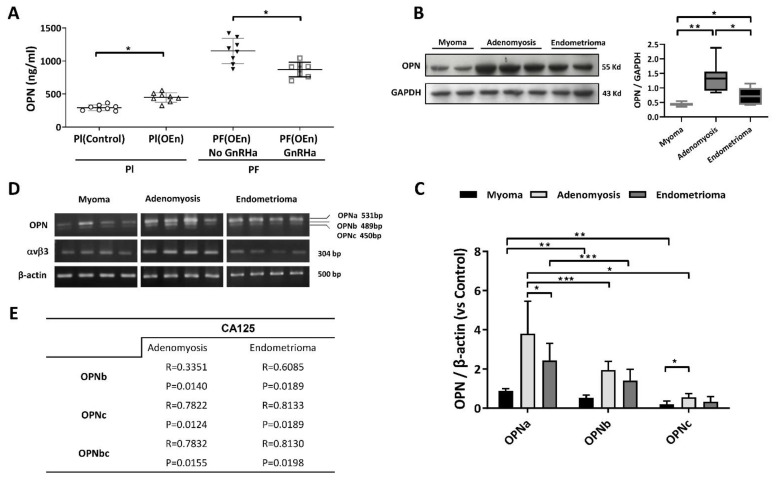
Expression of OPN and OPN isoforms in endometriotic lesions. (**A**) OPN levels in plasmas from women with myomas (n = 8, control) and women with ovarian endometrioma (n = 8), in peritoneal fluids from women with ovarian endometrioma (n = 8) before and after GnRH agonist treatment. (**B**) Representative blot image and quantitative analysis of Western blot data on OPN in myoma (n = 12), adenomyosis (n = 12), and ovarian endometrioma tissues (n = 12). The normalized OPN expression in adenomyosis was the highest, followed by the endometriosis tissues compared to the myoma tissues, *p* < 0.05. (**C**,**D**) Expressions of OPN isoforms (OPNa, OPNb, OPNc) in myoma, adenomyosis, and ovarian endometriosis tissues. All OPN isoforms were upregulated in endometriotic lesions, especially in adenomyotic tissue (*p* < 0.05). All OPN isoforms were expressed at significantly higher levels in endometriotic lesions than in the myoma control group (*p* < 0.05). (**E**) Linear correlation between CA125 and OPN isoforms in different endometriotic lesions. Plots represent the mean ± standard error of the mean (SEM). Pl: plasma; OEn: Ovarian endometrioma(s); PF: peritoneal fluid; GnRHa: gonadotropin-releasing hormone agonist; OPN: osteopontin; CA125: cancer antigen 125. (*) indicates a significant difference with *p* < 0.05. (**) indicates a significant difference with *p* < 0.01. (***) indicates a significant difference with *p* < 0.001.

**Figure 2 ijms-23-15328-f002:**
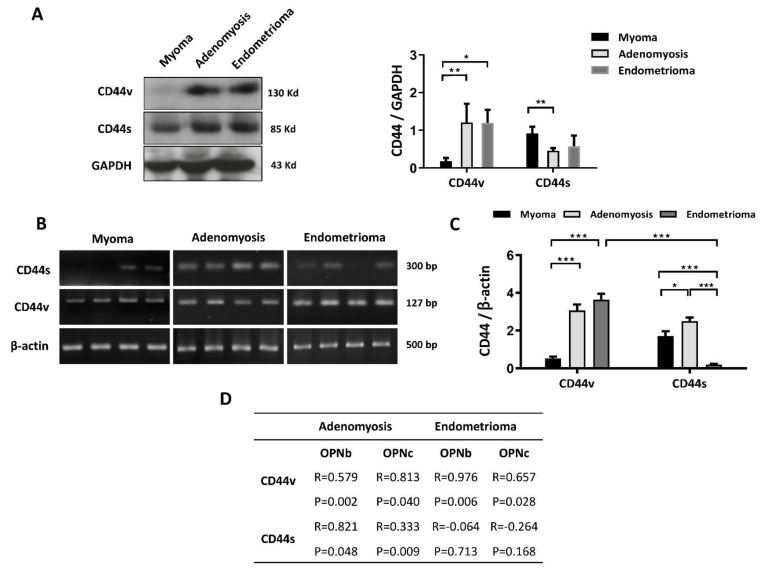
Differential expression of OPN ligand-receptor CD44s and its variant CD44v in endometriotic samples. (**A**) Quantification of proteomic data from controls and endometriotic tissues showed increased CD44v expression in endometriosis tissues. In contrast, both adenomyosis and ovarian endometriosis demonstrated slightly reduced CD44s expression. (**B**) The representative electrophoretogram of CD44s, CD44v, and β-actin in myoma, adenomyosis, and endometrioma is shown. (**C**) Normalized CD44 mRNA expression demonstrated the overexpression of CD44v in adenomyosis and ovarian endometrioma samples. In comparison to myoma cells, the CD44s variant was found to be overexpressed in adenomyosis cells but was diminished in endometrioma cells. (**D**) The linear correlation between the expression levels of CD44 variants and OPN isoforms in various endometriotic tissues. Plots represent the mean ± SEM. CD44s: CD44 standard; CD44v: CD44 variant. (*) indicates a significant difference with *p* < 0.05. (**) indicates a significant difference with *p* < 0.01. (***) indicates a significant difference with *p* < 0.001.

**Figure 3 ijms-23-15328-f003:**
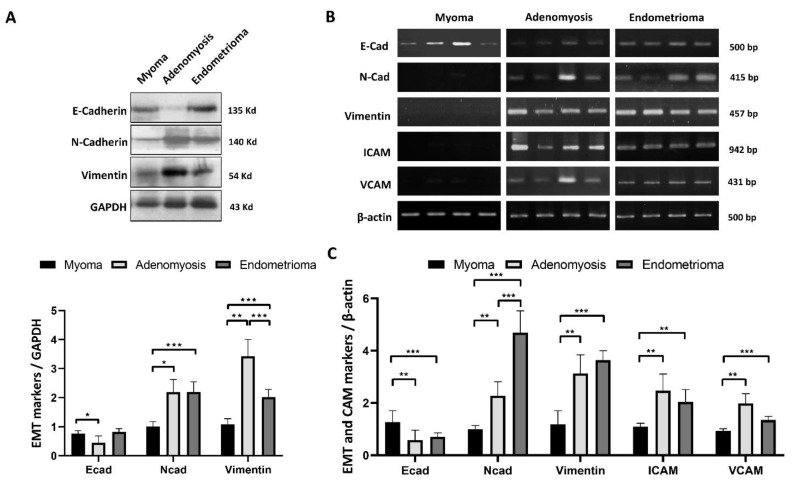
EMT markers and cell adhesion molecules (CAM) in myoma, adenomyosis and ectopic endometriosis lesions. (**A**) Representative chemiluminescence image of Western blot data demonstrated increased EMT markers in endometriosis. E-cadherin was underexpressed in adenomyotic lesions. N-cadherin and Vimentin were both found to be overexpressed in myoma tissue and endometriotic lesions. Additionally, the Vimentin concentration was extremely high in adenomyosis compared with endometrioma, myoma, and normal endometrium (*p* < 0.05). (**B**) The mRNA expression levels of EMT markers, including N-cadherin, E-cadherin, Vimentin, and CAM markers, ICAM and VCAM, were detected by quantitative real-time PCR, as shown in the electrophoretogram. (**C**) Analysis and normalization of the mRNA levels of EMT and CAM markers. E-cadherin was downregulated, while N-cadherin, Vimentin, ICAM and VCAM expression was upregulated in endometriotic lesions (*p* < 0.05). Plots represent the mean ± SEM. Ecad: E-cadherin; Ncad: N-cadherin; ICAM: intercellular adhesion molecule; VCAM: vascular cell adhesion molecule. (*) indicates a significant difference with *p* < 0.05. (**) indicates a significant difference with *p* < 0.01. (***) indicates a significant difference with *p* < 0.001.

**Figure 4 ijms-23-15328-f004:**
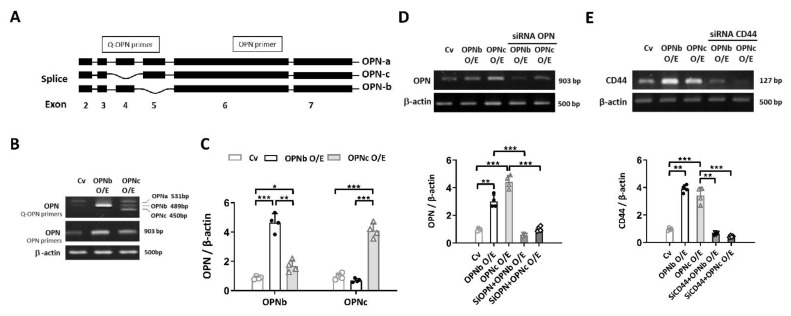
The efficiency of OPN isoform transfection and CD44 knockdown in OPN-overexpressing HEC1A cells was identified by real-time PCR. (**A**) Schematic illustration of OPN isoforms OPNa, OPNb, and OPNc. (**B**) OPN splice variants, OPNb and OPNc, were transfected into endometriotic cells. The vector only was used as a control (Cv). (**C**) The expression of OPN splice variants in the OPNb- or OPNc-transfected cell lines was identified by quantitative real-time PCR. (**D**) OPN silencing with OPN-siRNA in OPNb/OPNc-overexpressing endometriotic cells. (**E**) CD44 knockdown efficiency was identified by real-time PCR. Cv: control vector; O/E: overexpression. Plots represent the mean ± SEM, n = 4. (*) indicates a significant difference with *p* < 0.05. (**) indicates a significant difference with *p* < 0.01. (***) indicates a significant difference with *p* < 0.001.

**Figure 5 ijms-23-15328-f005:**
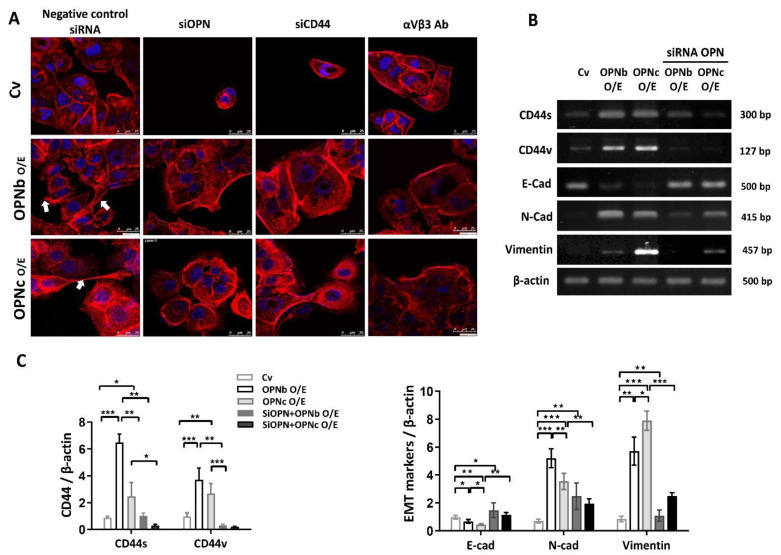
The effect of OPN on actin filament remodeling of HEC1A cells and the expression of OPN ligand receptors or EMT markers. (**A**) Phalloidin conjugates to the orange fluorescent dye phalloidin-tetramethylrhodamine (TRITC) for actin labeling and the blue fluorescent dye (DAPI) for nuclei and was observed under a Leica TCS SP5 confocal microscope. Overexpression of OPN splice variants induced pseudopodia extension. In contrast, the knockdown of OPN simultaneously reduced actin polymerization. The knockdown of CD44 or inhibition of αVβ3 integrin activity with the corresponding antibody concurrently reduces actin filament remodeling. (**B**) The PCR products of OPN ligand receptors (αvβ3, CD44s, and CD44v) and EMT markers (N-cad, E-cad, and Vimentin) in OPN-overexpressing (O/E) cells were observed using an electrophoretogram. (**C**) The mRNA expression levels of CD44v, CD44s, Vimentin, N-cadherin, and E-cadherin were detected by quantitative real-time PCR. Higher expression levels of CD44s and CD44v were found in OPN O/E cells. Additionally, the expression of CD44s in the OPNb-overexpressing cells was significantly higher than that in the other groups. Enhanced Vimentin and N-cadherin mRNA levels were found in conjunction with a reduced E-cadherin level in OPN O/E cells. Plots represent the mean ± SEM, n = 4. CD44s: CD44 standard; CD44v: CD44 variant; O/E: overexpression; si: siRNA; αVβ3 Ab: anti-αVβ3 integrin antibody; Ecad: E-cadherin; Ncad: N-cadherin. (*) indicates a significant difference with *p* < 0.05. (**) indicates a significant difference with *p* < 0.01. (***) indicates a significant difference with *p* < 0.001.

**Figure 6 ijms-23-15328-f006:**
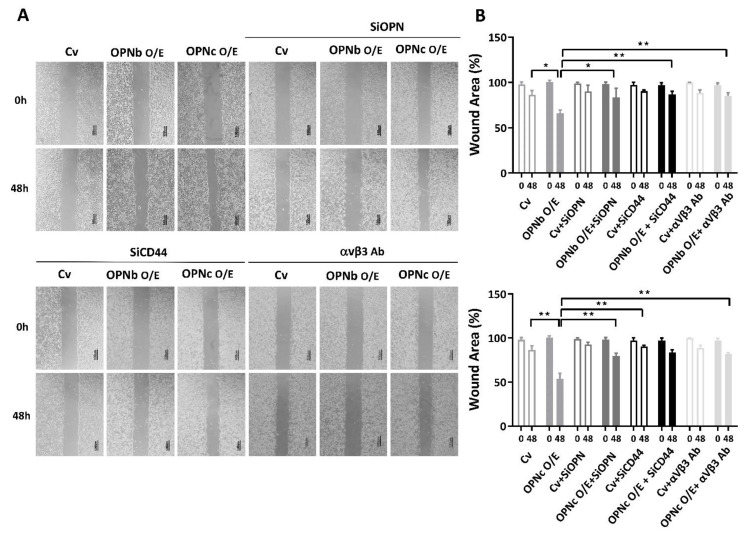
The effect of OPN isoforms on HEC1A cell migration. (**A**) Both OPNb and OPNc promoted cellular migratory ability, but the latter showed a significantly higher stimulation of endometriotic cell migration. Knockdown OPN or suppression of OPN ligand receptors (αvβ3 and CD44) reversed the changes in the wound healing area. (**B**) Plots show the differences in migration between control cells and OPN-overexpressing cells before and after suppression. Plots represent the mean ± SEM, n = 4. Cv: control vector, O/E: overexpression, si: siRNA, αVβ3 Ab: anti-αVβ3 integrin antibody. (*) indicates a significant difference with *p* < 0.05. (**) indicates a significant difference with *p* < 0.01.

**Figure 7 ijms-23-15328-f007:**
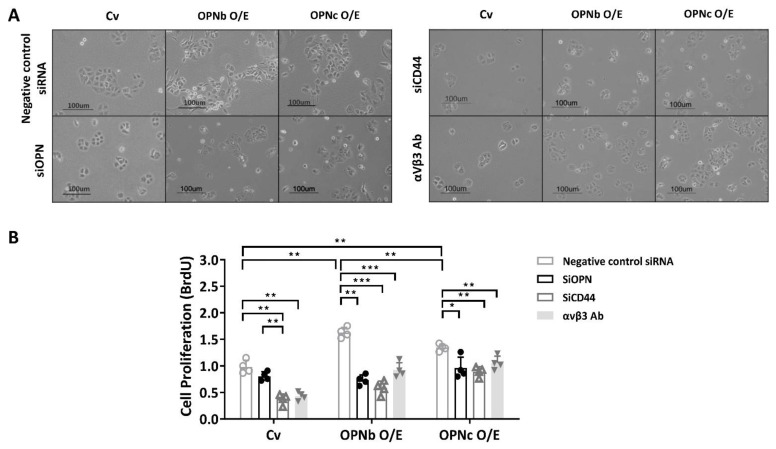
The effect of OPN on cell proliferation in OPN-overexpressing HEC1A cells. (**A**) The OPNb- or OPNc-overexpressing cells showed enhanced proliferation. The morphological change into a slender phenotype similar to the mesenchymal type was also found in the OPN-overexpressing cells. In contrast, through the depletion of OPN and OPN ligand receptors (CD44 and αvβ3), the cell phenotypes resembled those of the control cells. (**B**) The impact of OPN variants on cell proliferation was determined by using the BrdU Cell Proliferation Assay Kit. The results also showed that proliferation was more prominent in OPNb-overexpressing cells than in OPNc-overexpressing and control cells. Plots represent the mean ± SEM, n = 4. Cv: control vector; O/E: overexpression; si: siRNA, αVβ3 Ab: anti-αVβ3 integrin antibody. (*) indicates a significant difference with *p* < 0.05. (**) indicates a significant difference with *p* < 0.01. (***) indicates a significant difference with *p* < 0.001.

**Figure 8 ijms-23-15328-f008:**
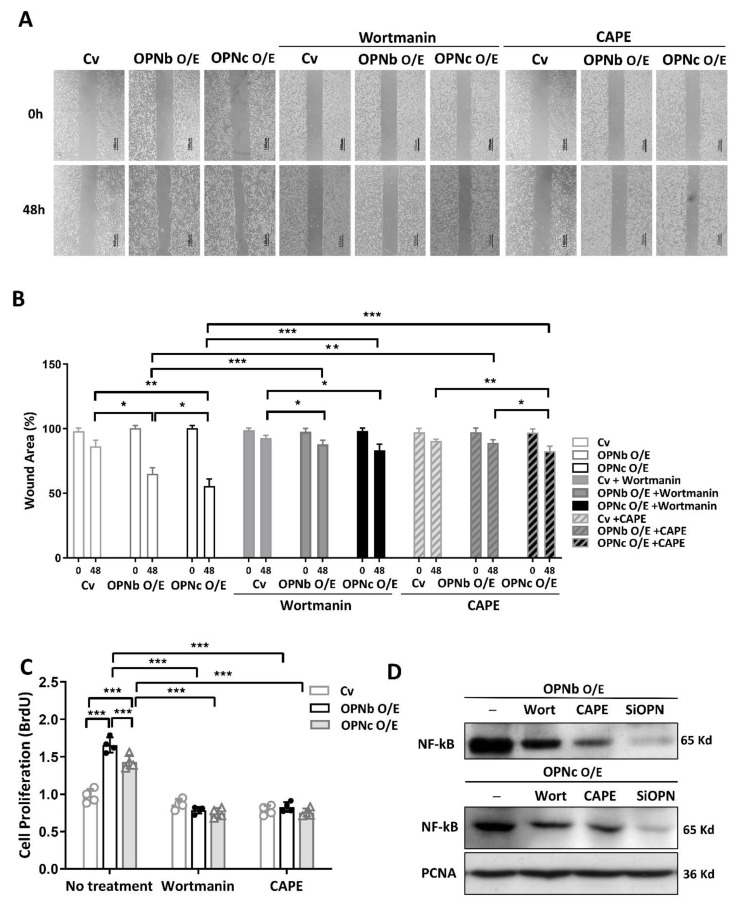
The effect of OPN variants on HEC1A cell migration via the PI3K or NF-κB signaling pathway. (**A**,**B**) A scratch wound assay was performed to investigate endometriotic cell migration. OPN isoforms can lead to cell migration, but treatment with 10 μM wortmannin (PI-3K inhibitor) or 10 μM CAPE (NF-κB inhibitor) for 24 h can slow down or suppress cell migration. (**C**) Wortmannin and CAPE inhibited proliferation in cells overexpressing OPN isoforms. (**D**) Nuclear translocation of NF-κB was inhibited by OPN suppression. Plots represent the mean ± SEM, n = 4. (*) indicates a significant difference with *p* < 0.05. (**) indicates a significant difference with *p* < 0.01. (***) indicates a significant difference with *p* < 0.001.

**Figure 9 ijms-23-15328-f009:**
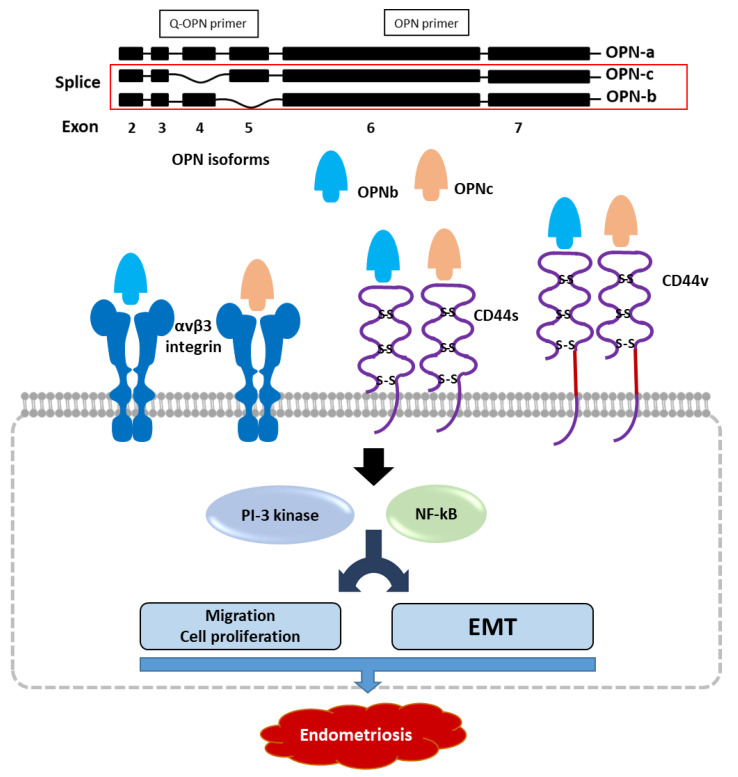
Schematic illustration of the role of OPN isoforms in endometriosis pathogenesis. OPNb and OPNc bind to OPN-ligand receptors CD44 and αvβ3 integrin, activating the PI3K and NFkB pathways, thereby inducing endometriotic cellular migration, proliferation, and the epithelial-mesenchymal transition process.

## Data Availability

The data presented in this study are available on request from the corresponding author. The data are not publicly available due to privacy.

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
