# Peer review of "Osteopontin Splicing Isoforms Contribute to Endometriotic Proliferation, Migration, and Epithelial-Mesenchymal Transition in Endometrial Epithelial Cells"

_ijms, 2022, doi:10.3390/ijms232315328_

Round 1
Reviewer 1 Report
The study by Ho and colleagues reports a possible causal relationship between the osteopontin variants and EMT related cellular functions in endometriotic cells. The authors showed the possible link between the two events are likely to be CD44v, integrins and the related molecular events. Overall, this is an interesting study and has addressed a medical problem. The design and data appear to be of good quality. There are however a number of issues to be address as stated in the following.
Results section 2.1 and Figure-1. The fluid samples were collected from 8 patients of each subgroup and tissues from 12 in each group. This is a small number for a major medical issue. What was the rationale for the selection of patient number, namely was this based on power prediction? Were different groups matched, in age, background condition and other clinical and underlying factors? It is necessary to clarify here or in the method section.
Figure-3C. The figure was said to demonstrate ‘Analysis and normalization of the mRNA levels of EMT markers’. Expression of EMT markers were presented in two separate panels (left and right). The two datasets are clearly different. What are the differences between these two bar graphs, other than that the right had two more factors (ICAM and VCAM) and why they are different (E-cadherin, N-cadherin and vimentin)?
Figure-5A. There are a number of issues with this figure. First: what are the images of the far left panel? In the CV group, the siOPN and siCD44 subgroups (top panel, middle two images), only a small number of cells (1 or 2) are shown compared with the rest of the images that a cluster healthy cells are shown. What was the reason for the selection of these cells: were the siOPN and siCD44 unhealthy? This needs to be addressed. It was said that pseudopodia was affected after modifications. It is highly advisable that these cellular structures are indicated, by arrows in the figure.
Figure-6 and Figure-7. From the methods, it seems that both cell migration and cell proliferation were tested over the same period, namely 48 hours. The pace of cell migration over 48 hours appears extremely and unusually slow. The changes of cell migration and cell proliferation also appear to be proportional by a quick estimate. Whilst it was very uncommon to see healthy living cells migrate this slow, would proliferation change contribute to the migration during the same period? Additionally, what cells are shown in these figures.
Figure-8: What cells were used and shown here?
Method. Section 4.3. It was stated that a cell line ‘HEC1A (ATCC: HTB-112TM) and human endometrial cells derived from clinical samples were used in this study’. However, the manuscript tends to describe the HEC1A cells without referring to the cells derived from clinical samples, the latter would be of particular interest. This point also needs to be clarified in the text, in results as well as in figure legends as appropriate.
Author Response
Thank you for your letter concerning our manuscript [ijms-2044282] entitled “Osteopontin Splicing Isoforms Contribute to Endometriotic Proliferation, Migration, and Endometrial-Mesenchymal Transition in Endometrial Epithelial Cells”, in which you asked us to make revisions in response to the your precious comments
In response to your comments, we make the following point-by-point responses as enclosed in the file

Reviewer 2 Report
This paper reports on OPN interacted with the CD44 receptor or integrin αvβ3 and the distinctive function of each OPN splicing variant in the pathogenesis of endometriosis. The results could promote the development of OPN-directed therapies for women with endometriosis and shed light on how endometriosis has negative effects on female reproduction. In the general, the authors have carried out an interesting and logical series of experiments, which I found to be quite interesting. However, this manuscript needs professional English editing to be published. There are some details that could use further attention.
1.Please provide three clear repetitions of the WB diagram in the article.
2.Please label the phenomenon in Figure 5A with a clear ruler
3.Standardize unit writing.
4.For example, in 434 lines μl to μL.
In 435 lines, ml to mL.
5.Add a method for observing PCR products by electrophoresis.
6.Reasonable labeling of saliency.
7.In the histogram vertical coordinate, 0.0 is changed to 0 and 1.0 to 1.
8.The author chose PI3K and NF-κB for the experiment, and suggested that other indicators of related pathways could be supplemented for more comprehensive and reliable exploration
Author Response
Thank you for your letter concerning our manuscript [ijms-2044282] entitled “Osteopontin Splicing Isoforms Contribute to Endometriotic Proliferation, Migration, and Endometrial-Mesenchymal Transition in Endometrial Epithelial Cells”, in which you asked us to make revisions in response to your comments
In response to the comments, we make the following point-by-point responses, in addition to a pdf file with blots and English editing cerfitication
Comments to the Author from Reviewer 2:
This paper reports on OPN interacted with the CD44 receptor or integrin αvβ3 and the distinctive function of each OPN splicing variant in the pathogenesis of endometriosis. The results could promote the development of OPN-directed therapies for women with endometriosis and shed light on how endometriosis has negative effects on female reproduction. In the general, the authors have carried out an interesting and logical series of experiments, which I found to be quite interesting. However, this manuscript needs professional English editing to be published. There are some details that could use further attention.
Response
We appreciated the precious opinion from the reviewer and made the following point-by-point responses.
Comment 1
Please provide three clear repetitions of the WB diagram in the article.
Response to comment 1
The three clear repetitions of WB diagram in the article were prepared as suggestion.
Comment 2
Please label the phenomenon in Figure 5A with a clear ruler.
Response to comment 2
We have already modified the manuscript according to your recommendations
Comment 3
Standardize unit writing. For example, in 434 lines μl to μL. In 435 lines, ml to mL.
Response to comment 3
We have already modified the manuscript according to your recommendations
Comment 4
Add a method for observing PCR products by electrophoresis.
Response to comment 4
We have already modified the manuscript according to your recommendations
Comment 5
Reasonable labeling of saliency.
Response to comment 5
We have already modified the manuscript according to your recommendations
Comment 6
In the histogram vertical coordinate, 0.0 is changed to 0 and 1.0 to 1.
Response to comment 6
We have already modified the manuscript according to your recommendations
Comment 7
The author chose PI3K and NF-κB for the experiment, and suggested that other indicators of related pathways could be supplemented for more comprehensive and reliable exploration.
Response to comment 7
Thanks for your suggestion. We added some indicators of the PI3K and NF-κB pathways in the Discussion Session.
From Line 392-Line 404, we added the revised sentence as followed:
By interacting with CD44 family of receptors, OPN can activate the cell survival (anti-apoptosis) signals through hospholipase C-γ (PLCγ–protein kinase C (PKC)–phosphatidylinositol 3-kinase (PI3K)–Akt pathway. OPN also exerts its function by activation of hypoxia-inducible factor-1 alpha (HIF-1α) pathways via the PI3k/AKT pathway leads to enhanced tumor cell survival, proliferation, invasion, EMT and angiogenesis. Upon binding to αvβ3, OPN can promotes cells motility and tumor progression through nuclear factor-inducing kinase (NIK)–ERK (extracellular signal-related kinase) and MEKK1 (mitogen-activated protein kinase kinase kinase1)–JNK1 (c-Jun N-terminal kinase 1) signaling pathways to active AP1. Concurrently, OPN can transactivate epidermal growth factor receptor (EGFR) to trigger ERK phosphorylation which eventually leads to activation of AP1. AP1 and NF-kB can activate both metalloproteinase (MMP) 2 and MMP9 to promote cell movement and metastasis.
